# Efficacy of Mesenchymal-Stromal-Cell-Derived Extracellular Vesicles in Ameliorating Cisplatin Nephrotoxicity, as Modeled Using Three-Dimensional, Gravity-Driven, Two-Layer Tubule-on-a-Chip (3D-MOTIVE Chip)

**DOI:** 10.3390/ijms242115726

**Published:** 2023-10-29

**Authors:** Eun-Jeong Kwon, Seong-Hye Hwang, Seungwan Seo, Jaesung Park, Seokwoo Park, Sejoong Kim

**Affiliations:** 1Department of Internal Medicine, Seoul National University Bundang Hospital, Seongnam 13620, Republic of Korea; dolce102003@gmail.com (E.-J.K.); jasno1@hanmail.net (S.-H.H.); 2Osong Medical Innovation Foundation, Cheongju-si 28161, Republic of Korea; seungwan@kbiohealth.kr; 3Department of School of Interdisciplinary Bioscience and Bioengineering, Pohang University of Science and Technology, Pohang 37673, Republic of Korea; jpark@postech.ac.kr; 4Department of Internal Medicine, Seoul National University College of Medicine, Seoul 03080, Republic of Korea

**Keywords:** extracellular vesicles (EVs), acute kidney injury, tubule-on-a-chip, 3D-MOTIVE chip, drug efficacy

## Abstract

Mesenchymal stromal cell (MSC)-derived extracellular vesicles (EVs) are known to have a therapeutic effect on nephrotoxicity. As animal models require significant time and resources to evaluate drug effects, there is a need for a new experimental technique that can accurately predict drug effects in humans. We evaluated the therapeutic effect of MSC-derived EVs in cisplatin nephrotoxicity using a three-dimensional, gravity-driven, two-layer tubule-on-a-chip (3D-MOTIVE chip). In the 3D-MOTIVE chip, 10 μM cisplatin decreased the number of attached cells compared to the vehicle. Conversely, annexin V and reactive oxygen species (ROS) were increased. Cell viability was increased 2.8-fold and 2.5-fold after treatment with EVs at 4 and 8 µg/mL, respectively, compared to the cisplatin-induced nephrotoxicity group. Cell attachment was increased 2.25-fold by treatment with 4 µg/mL EVs and 2.02-fold by 8 µg/mL EVs. Annexin V and ROS levels were decreased compared to those in the cisplatin-induced nephrotoxicity group. There were no significant differences in annexin V and ROS levels according to EV concentration. In sum, we created a cisplatin-induced nephrotoxicity model on a 3D-MOTIVE chip and found that MSC-derived EVs could restore cell viability. Thus, MSC-derived EVs may have the potential to ameliorate cisplatin-induced nephrotoxicity.

## 1. Introduction

Recently, a novel approach known as “cell-free therapy” has emerged in the field of kidney injury treatment [1,2]. This innovative therapeutic avenue capitalizes on extracellular vesicles (EVs), which are naturally released biocompatible particles that carry functionally active bioactive molecules. Cell-free therapy through EVs relies on the paracrine/endocrine mechanisms of stem cells and involves the administration of EVs in preclinical models of acute kidney injury (AKI) [3]. EVs, which contain various growth factors, mRNA, and DNA instead of cells, are known to exert effects through immunomodulatory, regenerative, and anti-inflammatory mechanisms in kidney injury models [4]. This groundbreaking technique has the potential to serve as a therapeutic intervention for cisplatin-induced nephropathy, harnessing the advantageous effects of EVs to facilitate renal repair and regeneration, without the need to directly administer stem cells. However, these fields have limitations in isolating extracellular vehicles (EVs) due to the challenges of homogenizing vesicles and low-volume extraction, which are also associated with high costs [5,6,7]. Our group successfully overcame these barriers by developing a flat-plate bioreactor in previous research [8]. This innovative bioreactor technology has facilitated the efficient large-scale isolation of EVs, rendering the process more cost-effective and enhancing the overall yield of isolated EVs for potential therapeutic applications. Our findings confirmed that EVs derived from MSCs decreased reactive oxygen species (ROS) and mitigated apoptosis in both in vivo and in vitro cisplatin-induced kidney injury models.

Despite demonstrating the efficacy of EVs in cisplatin-induced nephrotoxicity, our previous studies had some limitations. First, 2D cell culture systems do not represent a physiologic environment; these systems lack circulating blood and differ significantly from the conditions experienced by cells within actual organs. Additionally, human responses cannot always be accurately predicted based on data from animal studies, mainly because of inherent differences in physiology. Therefore, a preclinical 2D cell culture system and animal model are not suitable for effective drug screening, and systems that better mimic human organs are needed.

The organ-on-a-chip is a device that utilizes human cells and offers several advantages over using a 2D culture system or animal models. It is also a more relevant and human-specific model, maintaining a physiological microfluidic environment [9]. Fluid flow plays a pivotal role in maintaining the balance of organs and in morphogenesis and pathogenesis [10]. This flow creates both shear stress and pressure stress, which, in turn, influences cell proliferation, differentiation, and migration [11]. Additionally, it can evaluate the effectiveness of drug candidates at a low cost and in a short time.

In line with these considerations, we investigated the impact of MSC-derived EVs in a cisplatin-induced nephrotoxic model, using the tubular-on-a-chip device that incorporates kidney tubular and glomerular endothelial cells, the main targets of cisplatin-induced nephrotoxicity.

## 2. Results

### 2.1. MSC-EV Treatment Increases Cell Viability in a Cisplatin-Induced Nephrotoxicity Model

In the 3D-MOTIVE chip model, treatment with cisplatin caused cell detachment, and the effect was dose-dependent. Additionally, cisplatin treatment led to a decrease in TMRM fluorescence uptake, indicating an impact on mitochondrial stability (Appendix A). Specifically, after treatment with cisplatin at a concentration of 10 uM, cell viability was maintained at 50–70% compared to the control group.

In our study, we designed a three-dimensional, gravity-driven, two-layer tubule-on-a-chip system called a 3D-MOTIVE (Multi-Organ Tissue Chip, Interdisciplinary Value Escalation) chip, consisting of a single layer of polycarbonate (PC) material with three badge chambers and separate-fit inserts at each end of a single channel. Both sides of the insert membrane were coated with collagen type 1. We cultured primary human renal proximal tubule epithelial cells (RPTECs) on the bottom and primary human glomerular microvascular endothelial cells (GMVECs) on the top side of the insert device (Figure 1). We then transferred the insert device to the bottom plate of the 3D-MOTIVE chip with a bi-directional flow.

After the cells were treated with EVs at 4 and 8 µg/mL, cell viability was increased by 2.8-fold and 2.5-fold, respectively (Figure 2A,B), when compared to the cisplatin-induced nephrotoxicity group. Cell attachment was also increased 2.25-fold by 4 µg/mL EVs and 2.02-fold by 8 µg/mL EVs (Figure 2C). No statistically significant difference in cell viability or percentage of attached cells was found between the groups treated with EVs at 4 µg/mL or 8 µg/mL.

### 2.2. Tubular Function and Apoptosis in a Cisplatin-Induced Nephrotoxic Model Treated with MSC-Derived EVs

Cisplatin-associated renal injury first manifests when its metabolites enter the proximal tubule epithelial cells [12]. They are further metabolized to reactive thiol groups which are highly nephrotoxic. Clinically, serum levels of creatinine and albuminuria increase. Therefore, we evaluated tubular function according to the albumin uptake after EV treatment in a cisplatin-induced AKI model on a 3D-Motive chip. However, there was no effect of albumin uptake attributed to cisplatin and EVs (Figure 3A,B).

Apoptosis occurs in cisplatin-induced nephrotoxicity, and one of the mechanisms of this process is mitochondrial dysfunction [13]. The TMRM assay was conducted to monitor the mitochondrial potential. As depicted in Figure 3C, cisplatin tended to decrease TMRM intensity, which was not attenuated by EVs. However, in the detection of annexin V, the final stage of apoptosis, cisplatin caused a 1.5-fold increase, which was subsequently reduced to 0.5-fold by EVs compared to the control (Figure 3D). No differences were found between different EV treatment concentrations.

### 2.3. Protective Effect of Oxidative Stress of MSC-Derived EVs

Mitochondrial dysfunction plays a role in cisplatin-associated nephrotoxicity in the generation of oxidative stress, resulting in the accumulation of intracellular ROS [14]. Cisplatin caused a substantial 855-fold increase in ROS generation compared to the control (Figure 4). However, treatment with EVs significantly decreased ROS levels to the control level. Notably, the change in ROS levels was not influenced by the EV treatment concentration.

## 3. Discussion

This study assessed the protective effects of EVs in a model of cisplatin-induced nephrotoxicity, applying a three-dimensional, gravity-driven, two-layer tubule-on-a-chip model. EVs restored the viability of cisplatin-injured human primary kidney epithelial cells. The increase in cell viability observed with EVs was associated with apoptosis. Two primary mechanisms contributed to this protective effect. The first mechanism involved a significant reduction in the expression of annexin V mediated by EV treatment. The second mechanism was a decrease in ROS generation through antioxidative effects. However, these EV effects were not dose-dependent. The findings suggest that even small quantities of EVs were sufficient to confer a significant protective effect.

EVs are produced by nearly all cells and contain important biomolecules such as DNA, RNA, and proteins (e.g., growth factors) within the vesicle [3,15]. These materials are available in either natural or synthetic forms. They can function as therapeutic agents themselves or be incorporated into drug delivery systems. EVs are sourced from various cells, including mesenchymal stromal cells from bone marrow or placenta, dendritic cells, and platelets. We have conducted studies using EVs derived from human bone marrow mesenchymal stromal cells. However, the isolation, characterization, and standardization of EVs present significant technical challenges. These challenges encompass isolating EVs, ensuring the stability of the cell sources, optimizing the on-target effects of EVs, and minimizing off-target effects [7,16,17]. Representative methods used for the isolation of EVs include ultracentrifugation, ultrafiltration, precipitation, affinity isolation, chromatography, and density gradient centrifugation [7]. Although the first four methods are simple, the efficacy and purity of isolation are problematic, and the last three methods have the disadvantages of high cost and time requirements. Through previous research, we have overcome the limitations of EV isolation and produced high-yield EVs using a flat-plate bioreactor [8]. EVs are nanovesicles, and to be developed as clinical drugs in the future, their penetration into organs other than the target organ must be investigated. We are currently engaged in research aiming to identify substances with reno-protective effects, although these findings have not yet been published. The toxicity of EVs must also be studied. To secure this, our research team is developing a multi-organ chip and conducting research on the organ distribution of EVs using an animal model. We were able to demonstrate the on-target effects using a tubule-on-chip system and confirmed a dose effect for EVs. Currently, intravenous injection is required when using EVs clinically in humans, and the 4 ug/mL concentration used in our experiment is the target concentration in the blood. Thus, our study may provide guidance on EV concentrations in future clinical trials to increase epithelial cell viability.

In a prior study conducted by our research team, we induced cisplatin nephrotoxicity in HK-2 cells and subsequently treated them with EVs. In this experiment, we observed a significant decrease in the expression of ROS and NOX-4, along with an increase in the expression of PCNA. These findings suggest that the mechanism through which EVs exerted their effects in the current study may have involved both an antioxidant response and a positive impact on cell viability [8]. Previous reports suggested that miR-100-5-p found within EVs entered target cells and inhibited the expression of NOX-4 [18]. Additionally, EVs have been shown to carry antioxidant enzymes that can be directly employed as ROS scavengers [19]. The administration of EVs in animal experiments assessing the effectiveness of EVs in cisplatin-induced AKI resulted in decreased apoptosis [20]. Oxidative damage was decreased, as evidenced by lower expression of 8-OHdg. The histological analysis in another study confirmed that treatment with EVs reduced tubular damage and lymphocyte infiltration [21]. Nassar et al. studied the prevention of CKD progression through the anti-inflammatory effects exerted by directly injecting EVs into humans [22]. Changes were followed for 12 months after intravenous and intraarterial injections in patients with stages III-IV CKD. They demonstrated notable reductions in creatinine levels and urine albumin–creatinine ratios. Histological examination revealed substantial improvements in tubular epithelial cell damage. Although that study showed reno-protective effects using human EV data, it took an average of more than 8 weeks to show the drug efficacy of EVs. In other words, our study illustrates that MSC-derived EVs exhibit reno-protective properties in a cisplatin-induced nephrotoxicity model by enhancing cell viability through their antioxidant and anti-inflammatory effects.

The cisplatin-induced nephrotoxicity model under shear stress conditions in 3D-MOTIVE chip accurately replicated the in vivo situation. Fluidic shear stress on a chip is fundamental to mimicking the native renal tubule environment [23] as it helps the polarization and localization of the tight junction of tubular epithelial cells. Although our research did not definitively identify the role of endothelial cells, based on previous studies, the importance of the co-culture system and fluidic condition can be recognized [24,25]. Tubular cells exhibit distinct characteristics in the apical and basolateral membranes to facilitate the exchange of substances between the blood and the lumen. Consequently, the expression of transporters varies accordingly. This phenomenon occurred through co-culture with fluidic shear stress, and a significant up-regulation in gene expression related to transporters was observed [26,27].

Organ-on-a-chip technology offers significant advantages over traditional preclinical animal models like mice or dogs. One key benefit is derived from using human primary cells, providing a highly relevant and accurate representation of human biology. In vivo models offer the advantage of observing functional changes in the entire kidney, but they have limitations in examining kidney structures in fine detail. Conversely, organ-on-a-chip technology has the flexibility to simulate either a distal tubule or glomerulus on the chip, depending on the specific research objectives. Additionally, assessing drug efficacy in just 4 days is a considerable time-saving advantage, leading to quicker decisions in drug development. This accelerated timeline substantially decreases the time and resources needed for preclinical testing, ultimately expediting the development of potential therapies. The 3D-MOTIVE chip simulated an acute cisplatin-induced nephrotoxicity model, as depicted in the experimental design shown in Figure 1. 

In contrast to prior studies, the 3D-MOTIVE chip developed by our research team presents a distinct advantage by combining the benefits of both 2D cell culture systems and animal experiments. This advancement not only validates apoptosis, as demonstrated in previous animal experiments, but also enables the verification of cell functionality through assessments such as albumin uptake and mitochondrial stability. Additionally, the 3D-Motive chip allows the rapid evaluation of drug effectiveness in treating tubular cell injuries. The incidence of nephrotoxicity caused by cisplatin was reported as between 20% and 30%. Cisplatin is known to accumulate in tubular cells, leading to tubular cell injury, inflammation, and cell death, ultimately resulting in a reduction in tubular function [28]. Conducting clinical trials to validate the efficacy of EV treatment for cisplatin-induced nephrotoxicity in cancer patients presents practical challenges. These trials involve subjects with heterogeneous characteristics, such as variations in weight, ongoing chemotherapy, and medication adjustments based on clinical status. Most significantly, it is practically unfeasible to conduct kidney biopsies both before and after treatment to confirm the reno-protective effects attributed to EVs. Consequently, an alternative solution lies in the 3D MOTIVE chip, which not only minimizes time consumption but also enables the monitoring of cellular changes. While these advantages exist, it does have limitations in observing long-term changes and recovery. Previous research on cisplatin-induced nephrotoxicity has demonstrated that tubular injury marker NGAL changes occur earlier than the urine albumin–creatinine ratio [29]. It has been reported that alterations in renal function become apparent several months after cisplatin administration, indicating that cellular damage precedes functional impairment. These points are considered to be evidence that changes in TMRM and albumin uptake could not be confirmed in our study. Functional impairment was not evident due to cisplatin damage for 24 h. A longer-duration model may be needed to confirm functional impairment.

Nephrotoxicity has been reported in humans at cisplatin doses of 20 mg/m² or higher [30,31]. In a single-center prospective study investigating the relationship between dosage and nephrotoxicity, plasma platinum concentration exceeded 2 µg/mL at a mean infusion dose of 53.5 ± 19 mg/m², leading to the emergence of subclinical kidney injury [32]. When comparing cisplatin toxicity in 2D culture models to that in organ-on-a-chip models, it is noteworthy that the chip’s physiological conditions closely resemble those of humans. Although three times the dose is required to induce cisplatin injury in the organ-on-a-chip model, the induced concentration is nearly identical to the patient’s data [8].

This study had some limitations. Future studies should establish effective substances with reno-protective effects in EVs and investigate their specificity for tubular cells through molecular and gene studies. Additionally, pathways involved in the effects of EVs on cisplatin-induced nephrotoxicity should be studied at the molecular level. Furthermore, a detailed analysis is necessary to understand the specific impact of endothelial cells on the function and structure of tubules within the tubular-on-chip system, especially in the context of flow and co-culture systems. While we observed damage and epithelial cell function in proximal tubule cells using the 3D-MOTIVE chip, we did not observe distal tubules or glomeruli due to limitations in our ability to simultaneously view other sectors of the same organ. Lastly, we need to evaluate how many days the effect can be seen for, using the 3D-MOTIVE chip.

## 4. Materials and Methods

### 4.1. Cell Culture

Primary human renal proximal tubule epithelial cells (RPTECs; CC-2553) were obtained from Lonza (Basel, Switzerland). Primary human glomerular microvascular endothelial cells (GMVECs; ACBRI 128) were obtained from Cell Systems. Human RPTECs were cultured in renal epithelial growth medium (REGM Bllekit; CC-3191, Lonza, Basel, Switzerland) and used within 5–7 passages. Primary human GMVECs were cultured with Complete Classic Medium supplemented with serum and culture Boost TM (Cat. # 4Z0-500, Cell Systems) and were used within passage numbers 7–9. The cells were grown to > 90% confluence in T-175-flasks, which was reproduced at a concentration of 3 × 10^6^ cells/mL. The media were replaced once every 1–2 days.

### 4.2. Tubular-on-a-Chip in 3D-Motive Chip and Cisplatin-Induced Nephrotoxicity Model

The 3D-MOTIVE chip was made according to the schematic diagram shown in Figure 1. Injection molding was used to fabricate the chip, to mimic the renal environment; this is a reproducible manufacturing process to provide the consistency of the in vitro organ model [33]. The 3D-MOTIVE chip consists of a single layer of polycarbonate (PC, SABIC Lexan^®^ 121R) material with 3 badge chambers and separate-fit inserts at each end of a single channel. PC was selected as the material used to fabricate the chip, as it had low cost, high impact resistance, low moisture absorption, and excellent processing characteristics [34]. We performed a biocompatibility test using mouse NCTC clone 929 cells (L-929, Cat# CCL-1, ATCC) according to ISO 10993-5 [35] and confirmed that there was no cytotoxicity. The bottom of the motive chip and the surrounding areas on all 4 sides provide channels for the flow of the medium. Double-sided tape (3 M, Saint Paul, MN, USA) was used to secure the main body of the chip to prevent any leakage or loss of medium. After the chip body was disinfected with 70% ethanol and the bottom of the 3D-MOTIVE chip was adhered to double-sided tape, it was sterilized for cell culture using UV irradiation for 1 h on a clean bench. The chip was then prepared for filling the cell culture medium in each reservoir. A chip consisted of the main microfluidic chip with channels and a separatable insert module for cell culture. ThinCert^®^ (Cat. #662641, Greiner bio-one, Frickenhausen, Germany) with a pore diameter of 0.4 μm was used for the insert. Both sides of the insert membrane were coated with 100 μL (100 µg/mL) of collagen type 1 at 37 °C for 1 h to create an extracellular matrix. As the chip is not an integrated structure but a separable insert-type structure, it was placed oppositely so that the membrane could go upward before mounting the insert on the chip. We cultured RPTECs on the bottom and GMVECs on the top side of the insert device (Figure 1). RPTECs were first seeded at 3 × 10^5^ on the lower side of the membrane for 2 h, and then the insert was placed upright in the plate, and GMVEC were seeded on the inside of the insert membrane at 3 × 10^6^. GMVEC was cultured on the upper side of RPTEC under the membrane. The flow was directed to the RPTEC side.

We designed the flow to the 3D-MOTIVE chip at an angle of 7° and intervals of 8 min using a rocker that exerted a force equivalent to 0.13 dyne/cm^2^ at 37 °C in a humidified 5% CO_2_ atmosphere. The flow was directed to the RPTEC side. After 24 h of inoculation, the cells were stimulated with cisplatin (DongAh, Republic of Korea) at 10 μM for 24 h. After replacing the medium, the cells were treated with 4 µg/mL and 8 µg/mL of EVs for 24 h.

### 4.3. Extracellular Vesicle Collection

EVs extracted from previous studies [6] were used. Briefly, in bioreactor condition, hBM-MSCs purchased from the Catholic Institute of Cell Therapy (CIC, Seoul, Republic of Korea) were seeded at 1 × 106 and grown in 20% fetal bovine serum (FBS) (Cat. # 12 483 020, Gibco) for 3 days. The medium was then replaced with Dulbecco’s modified Eagle medium containing 10% FBS and cultured for 24 h in a humidified incubator for EV preparation. EVs were then separated from FBS using ultracentrifugation at 100,000× *g* and 4°C for 20 h.

EVs were prepared by centrifuging at 500× *g* for 10 min to remove detached cells. The supernatants were centrifuged at 3000× *g* for 20 min to remove cellular debris, and EVs were separated by density gradient ultracentrifugation. The 10–25% density gradient fraction was then collected. Subsequently, these collected samples were stored at −80 °C until use.

### 4.4. Cell Viability Assay (Calcein AM)

For the cell viability assay, cells were incubated with 2 μM calcein acetoxymethyl ester (calcein AM; Cat. #C3099, Invitrogen, Waltham, MA, USA) and Hoechst33342 (trihydrochloride, trihydrate, H3570) at 37 °C for 30 min. Subsequently, they were washed with phosphate-buffered saline. Fluorescent-labeled marker images were acquired with a Zeiss LSM 800 confocal microscope using a (×20) oil immersion objective lens. Mean fluorescence intensity (MFI) was measured using a fluorescence intensity analyzer.

### 4.5. Albumin Endocytosis

After treatment with EVs, the cells were washed with PBS and incubated with FITC-albumin (A9771, 1:10, Sigma-Aldrich, Saint Louis, MO, USA) at 37 °C for 30 min, followed by rinsing with cold PBS (4 °C). Subsequently, the cells were fixed with 4% paraformaldehyde at 25 °C for 15 min, permeabilized with 0.1% Triton X-100 for 2 min, and blocked with 3% nonfat milk in PBS for 30 min. The cells were then incubated with a specific human primary antibody, ZO-1 (ab221547, 1:10, Abcam, Cambridge, UK), at 4 °C overnight to stain the plasma membrane. Fluorescent-labeled marker images were acquired with a confocal microscope (×20) to quantify albumin endocytosis. MFI was measured using a fluorescence intensity analyzer.

### 4.6. Estimation of Mitochondrial Membrane Potential

The cells were treated with tetramethyl rhodamine methyl ester (TMRM; T668, Thermo Fisher Scientific, Waltham, MA, USA) at a final concentration of 200 nM at 37 °C for 30 min. RPTECs were then washed 3 times with PBS and counterstained with Hoechst33342. A confocal microscope was used to visualize fluorescence location. Images were taken with a ZEN 2 and captured at a 20× magnification.

### 4.7. Detection of ROS Generation

Intracellular ROS levels were assessed in the tubule-on-a-chip study using a DCFDA/H2DCFDA Cellular ROS assay kit. Following treatment with cisplatin with or without EVs, the cells were exposed to 20 μM DCFDA in 2 mL of media for 45 min at 37 °C in the dark. Confocal microscopy was used to visualize intracellular ROS levels.

### 4.8. FITC-Labeled Annexin V Staining

For the apoptosis assay, cells were stained with 100 μL of fluorescein isothiocyanate (FITC)-labeled annexin V (ab14085, Abcam) in 2 mL of media for 10 min and counterstained with Hoechst33342. The cells were then washed with annexin V binding buffer (0.1 M Hepes (pH 7.4), 1.4 M NaCl, and 25 mM CaCl2 solution). The staining process was performed according to the manufacturer’s instructions. The results were analyzed using fluorescence microscopy. The wavelength was set in the range of 480–520 nm for measurements.

### 4.9. Statistical Analysis

The SPSS statistical software package 18.0 (SPSS, Inc., Chicago, IL, USA) and GraphPad prism 8.0.1 (La Jolla, CA, USA) were used to perform all statistical analyses. Data are presented as means ± standard error (SE) and were analyzed using Student’s *t*-test or one-way ANOVA if normality was satisfied according to the Shapiro–Wilk test. If normality was not satisfied, the data were analyzed using the Mann–Whitney U test to compare 2 groups or the Kruskal–Wallis test to compare 3 or more independent groups. One-way analysis of variance followed by Dunnett’s multiple-comparison test was applied for multiple comparisons. *p*-values of less than 0.05 were considered statistically significant.

## 5. Conclusions

We successfully established a model of cisplatin-induced nephrotoxicity on the 3D-MOTIVE chip, demonstrating that EVs derived from MSCs could significantly improve cell viability. This suggests that MSC-derived EVs hold promise as a potential intervention for mitigating cisplatin-induced nephrotoxicity.

## Figures and Tables

**Figure 1 ijms-24-15726-f001:**
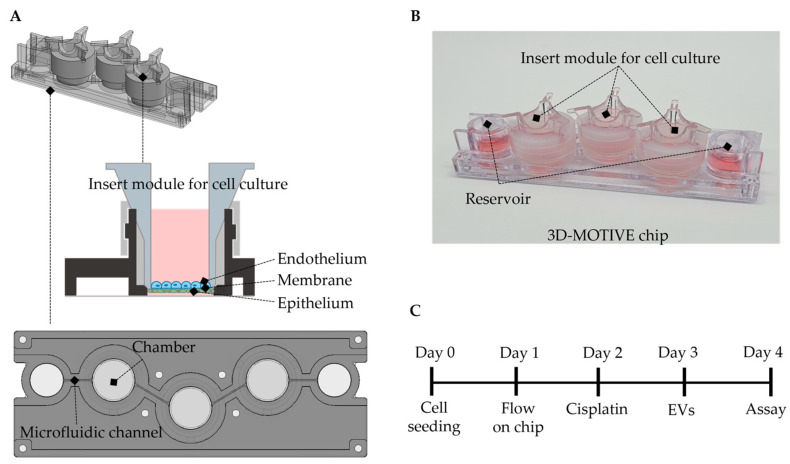
Schematic illustration of the experimental tools and schedule. (**A**) Design of the 3-dimensional, gravity-driven, 2-layer tubule-on-a-chip. (**B**) Actual image of a 3D-MOTIVE chip. (**C**) Cisplatin nephrotoxicity experimental schedule for 4 days using a 3D-MOTIVE chip.

**Figure 2 ijms-24-15726-f002:**
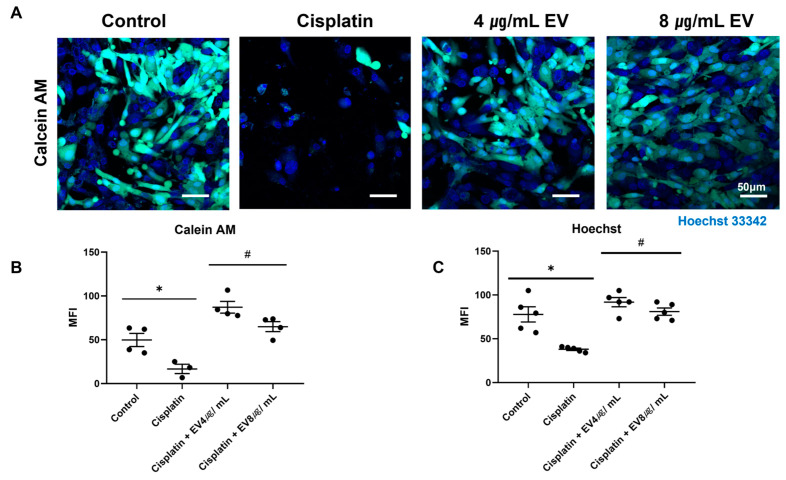
Cell viability of MSC-derived EVs in cisplatin-induced nephrotoxic model. (**A**) Confocal microscopic images (20×) of cells treated with a combination of cisplatin (10 μM) and EVs (4 µg/mL or 8 µg/mL) followed by staining with calcein AM and Hoechst33342. Scale bar, 50 μM. (**B**,**C**). Fluorescence intensity was measured (* *p* < 0.05 vs. control in Student’s *t*-test, # *p* < 0.05 vs. cisplatin in one-way ANOVA, data are presented as mean ± SE; *n* = 4–5).

**Figure 3 ijms-24-15726-f003:**
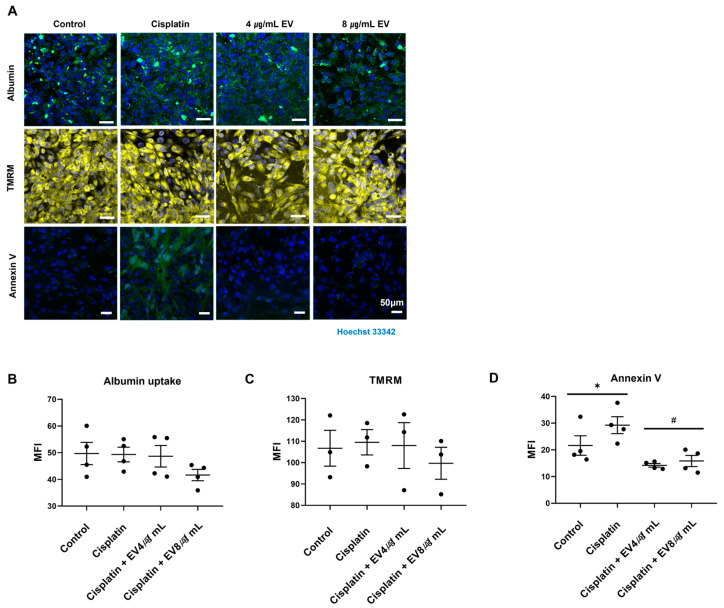
Tubular function and apoptosis in the cisplatin-induced nephrotoxic model by MSC-derived EVs. (**A**) Confocal microscopy images (20×) of cells treated with a combination of cisplatin and Evs followed by staining with calcein AM, TMRM, annexin V, and Hoechst 33342. Fluorescent dye scale bar, 50 μM. (**B**–**D**) Fluorescence intensity was measured (* *p* < 0.05 vs. control in Student’s *t*-test, # *p* < 0.05 vs. cisplatin in one-way ANOVA, data are presented as mean ± SE; *n* = 3–5).

**Figure 4 ijms-24-15726-f004:**
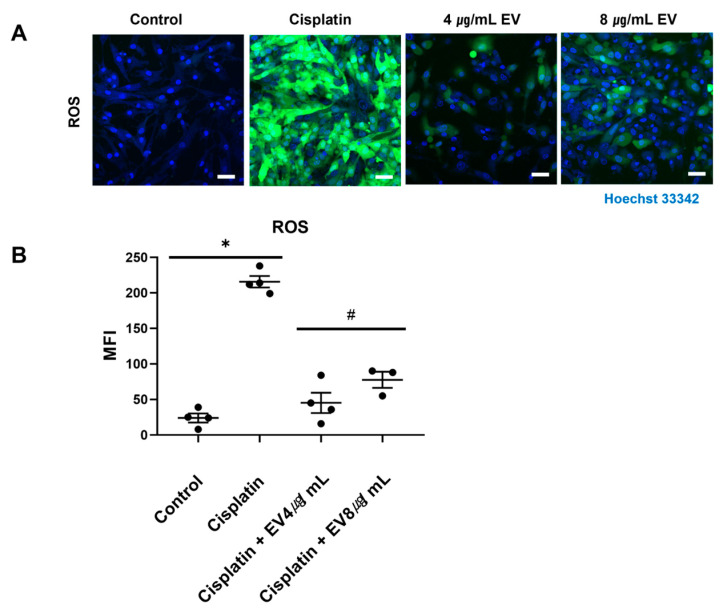
ROS generation of MSC-derived EVs. (**A**) Confocal microscopy images (20×) of cells treated with a combination of cisplatin and EVs, followed by incubation with DCF-DA, and staining with Hoechst 33342. Fluorescent dye scale bar, 50 μM. (**B**) Fluorescence intensity was measured (* *p* < 0.05 vs. control in Student’s *t*-test, # *p* < 0.05 vs. cisplatin in one-way ANOVA, data are presented as mean ± SE; *n* = 4).

## Data Availability

Data will be made available upon reasonable request.

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
