# Peer review of "Efficacy of Mesenchymal-Stromal-Cell-Derived Extracellular Vesicles in Ameliorating Cisplatin Nephrotoxicity, as Modeled Using Three-Dimensional, Gravity-Driven, Two-Layer Tubule-on-a-Chip (3D-MOTIVE Chip)"

_ijms, 2023, doi:10.3390/ijms242115726_

Round 1

Reviewer 1 Report

Comments and Suggestions for Authors

In the article titled 'Efficacy of mesenchymal stem cell-derived extracellular vesicles in cisplatin nephrotoxicity using a three-dimensional gravity-driven two-layer tubule-on-a-chip (3D-MOTIVE chip),' Eun-Jeong Kwon and colleagues describe a novel 3D system that enables a more rapid assessment of the effectiveness of EVs isolated from MSCs in the treatment of cisplatin-induced nephrotoxicity

·       First and foremost, the authors should change the definition from 'mesenchymal stem cells' to 'mesenchymal stromal cells,' which is nowadays considered more accurate by those working in the field of mesenchymals, as the characteristics of the latter are not entirely comparable to those of stem cells.

·       In the introduction, at line 38-39, "However, it has encountered a limitation in isolating mesenchymal stem cell (MSC)-derived extracellular vehicles (EVs) due to challenges of homogenizing vesicles and low-volume extraction, which can also incur a high cost." The authors should clarify what they mean by this statement. Cost and isolation issues related to MSCs? It depends on the sources. Those from bone marrow pose a risk to the donor, require extended expansion times with a risk of senescence (10.1186/s13287-023-03352-1); those isolated from placental tissues, such as the amniotic membrane, do not pose a risk in terms of collection as the material is obtained at delivery. They allow for rapid isolation of a large number of cells, reducing the risk of senescence during the process (10.1186/s13287-021-02607-z).

·       Check if the symbols for microliters (μL) and milliliters (mL) conform stylistically to the journal's requirements.

·       The title of the initial results should be revised and modified, for example, "MSC-EV Treatment Increases Cell Viability in a Cisplatin-Induced Nephrotoxicity Model."

·       The authors place significant emphasis on the 3D model in the introduction. However, considering that the entire discussion revolves around the therapeutic effect of EVs, they should strike a better balance. This could involve expanding the introduction section to explain why and how MSC-derived EVs could be effective.

·       Please provide a clear and explicit description of the methodology for the preparation of extracellular vesicles (EVs). While I understand the reference to a previous publication, if the EVs are derived from cells cultured in fetal bovine serum (FBS), this should be directly stated.

·       The increase in albumin observed in the IF at a concentration of 4 µg/ml EV-MSC, can it be attributed to an artifact? Because this image is not representative of the data in the histograms

·       Did the authors verify if the distribution of the results was normalized?

Comments on the Quality of English Language

Must be improved

Author Response

[23-Oct-2023]

Dear Reviewer1.

We would like to thank you for the letter dated 05/10/2023, and the opportunity to resubmit a revised copy of this manuscript. We would also like to take this opportunity to express our thanks to the reviewers for the positive feedback and helpful comments for correction or modification. We believe have resulted in an improved revised manuscript, which you will find uploaded alongside this document. The manuscript has been revised to address the reviewer's comments, which are appended alongside our responses to this letter.

Reviewer 1.

In the article titled 'Efficacy of mesenchymal stem cell-derived extracellular vesicles in cisplatin nephrotoxicity using a three-dimensional gravity-driven two-layer tubule-on-a-chip (3D-MOTIVE chip),' Eun-Jeong Kwon and colleagues describe a novel 3D system that enables a more rapid assessment of the effectiveness of EVs isolated from MSCs in the treatment of cisplatin-induced nephrotoxicity

Q1·       First and foremost, the authors should change the definition from 'mesenchymal stem cells' to 'mesenchymal stromal cells,' which is nowadays considered more accurate by those working in the field of mesenchymals, as the characteristics of the latter are not entirely comparable to those of stem cells.

A1.Thank you for your suggestion. The International Society for Cell & Gene Therapy (ISCT®) Mesenchymal Stromal Cell (ISCT MSC) committee also recommends clarifying the nomenclature of the term, so we will change “mesenchymal stem cells” to “mesenchymal stromal cells.”

Q2·       In the introduction, at line 38-39, "However, it has encountered a limitation in isolating mesenchymal stem cell (MSC)-derived extracellular vehicles (EVs) due to challenges of homogenizing vesicles and low-volume extraction, which can also incur a high cost." The authors should clarify what they mean by this statement. Cost and isolation issues related to MSCs? It depends on the sources. Those from bone marrow pose a risk to the donor, require extended expansion times with a risk of senescence (10.1186/s13287-023-03352-1); those isolated from placental tissues, such as the amniotic membrane, do not pose a risk in terms of collection as the material is obtained at delivery. They allow for rapid isolation of a large number of cells, reducing the risk of senescence during the process (10.1186/s13287-021-02607-z).

A1. We apologize for any confusion the reader may have in understanding the text. The part I mentioned describes the limitations of the technology itself for isolating extracellular vesicles (EVs). Review papers in the references also point out limitations in equipment for extracting EVs, especially using ultracentrifugation equipment and limitations in extracting insufficient amounts of EVs for clinical use.

Therefore, I changed line 41 of the Introduction section as follows.

However, these fields have limitations in isolating extracellular vehicles (EVs) due to the challenges of homogenizing vesicles and low-volume extraction, which are also associated with high costs

I also described this in line 157 of the Discussion section.

“Representative methods used for the isolation of EVs include ultracentrifugation, ultrafiltration, precipitation, affinity isolation, chromatography, and density gradient centrifugation [7]. Although the first 4 methods are simple, the efficacy and purity of isolation are problematic, and the last 3 methods have the disadvantages of high cost and time requirements. “

Q3·       Check if the symbols for microliters (μL) and milliliters (mL) conform stylistically to the journal's requirements.

A3. Thanks for your advice. As recommended by the IJMS, the liter notation for microliters and milliliters was changed to uppercase.

Q4·       The title of the initial results should be revised and modified, for example, "MSC-EV Treatment Increases Cell Viability in a Cisplatin-Induced Nephrotoxicity Model."

A4. Thanks for your suggestion. The title of the first result was modified as follows, "MSC-EV Treatment Increases Cell Viability in a Cisplatin-Induced Nephrotoxicity Model."

Q5·       The authors place significant emphasis on the 3D model in the introduction. However, considering that the entire discussion revolves around the therapeutic effect of EVs, they should strike a better balance. This could involve expanding the introduction section to explain why and how MSC-derived EVs could be effective.

A5. Thank you for your suggestion.

In line 32 of the Introduction section, I described EVs and their known mechanisms in kidney injury.

“Extracellular vesicles (EVs) are naturally released biocompatible particles that carry functionally active bioactive molecules. Cell-free therapy through EVs relies on the paracrine/endocrine mechanisms of stem cells and involves the administration of EVs in preclinical models of acute kidney injury (AKI) [3]. EVs, which contain various growth factors, mRNA, and DNA instead of cells, are known to exert effects through immunomodulatory, regenerative, and anti-inflammatory mechanisms in kidney injury models [4].

Q6·       Please provide a clear and explicit description of the methodology for the preparation of extracellular vesicles (EVs). While I understand the reference to a previous publication, if the EVs are derived from cells cultured in fetal bovine serum (FBS), this should be directly stated.

A6. I agree with your point.

The hBM-MSC culture and the EV separation method were independently discussed when describing the collection of EVs in line 306 in the Methods section. It was also clarified that during the culture of MSCs, a 2-step approach was employed, with 20% FBS used for the initial 3 days, followed by a transition to 10% FBS for the subsequent 1-day period.

Q7·       The increase in albumin observed in the IF at a concentration of 4 µg/ml EV-MSC, can it be attributed to an artifact? Because this image is not representative of the data in the histograms

A7. I agree with your point. The subsequently reevaluated statistical values did not yield significant results for either albumin uptake or TMRM in either the cisplatin group or the EV-treated group. The reason why it did not have a significant effect on albumin uptake was further discussed in line 218 and 236 of the Discussion section as follows.

“The 3D-MOTIVE chip simulated an acute cisplatin-induced nephrotoxicity model, as depicted in the experimental design shown in Figure 1.“

“While these advantages, it does have limitations observing long-term changes and recovery. Previous research on cisplatin-induced nephrotoxicity has demonstrated that tubular injury marker NGAL changes occur earlier than the urine albumin creatinine ratio [29] It has been reported that alterations in renal function become apparent several months after cisplatin administration, indicating that cellular damage precedes functional impairment. These points are considered to be evidence that changes in TMRM and albumin uptake could not be confirmed in our study. Functional impairment was not evident due to cisplatin damage for 24 hours. A longer duration model may be needed to confirm functional impairment.”

Q8·       Did the authors verify if the distribution of the results was normalized?

A8. I agree with your point. All data were checked for normality through the Shapiro-Wilk test before performing statistical analysis, and p > 0.05 was satisfied.

This content is described in line 351 of the Methods section.

“Data are presented as means ± standard error (SE) and were analyzed using the student t-test or one way ANOVA if normality was satisfied according to the Shapiro-Wilk test. If normality was not satisfied, the data were analyzed using Mann-Whitney U-test to compare 2 groups or the Kruskal-Wallis test to compare 3 or more independent groups. One-way analysis of variance followed by Dunnett’s multiple-comparison test, was applied for multiple comparisons.”

We very much hope the revised manuscript is accepted for publication in the Journal.

Thank you for your consideration. I look forward to hearing from you.

Reviewer 2 Report

Comments and Suggestions for Authors

The study entitled Efficacy of mesenchymal stem cells-derived extracellular vesicles in cisplatin nephrotoxicity using three-dimensional gravity-driven two-layer tubule-on-a-chip (3D-MOTIVE chip) addresses an important and clinically relevant issue: the evaluation of mesenchymal stem cells (MSCs)-derived extracellular vesicles (EVs) in cisplatin nephrotoxicity. The use of a three-dimensional gravity-driven two-layer tubule-on-a-chip (3D-MOTIVE chip) model provides a novel and potentially more clinically relevant experimental platform compared to traditional animal models.

1.       The use of the 3D-MOTIVE chip is commendable for its potential to mimic the in vivo renal environment more closely. However, it would be helpful to provide more details about the design and fabrication of the chip, as well as its validation and reproducibility. This information would enhance the transparency and credibility of the experimental model.

2.       The study reports improvements in cell viability, cell attachment, and reductions in Annexin V and ROS levels after EV treatment in the cisplatin-induced nephrotoxicity model. While these results are promising, it would be beneficial to include a more detailed discussion of the mechanistic insights into how MSCs-derived EVs are exerting their therapeutic effects. Providing insights into the potential signaling pathways or molecular mechanisms involved would enhance the paper's quality.

3.       It's crucial to include more information regarding the statistical methods used and the statistical significance of the observed differences. Clearly state the p-values and the statistical tests applied to determine significance. This will help readers assess the reliability of the findings.

4.       The paper mentions that increases in TMRM and albumin uptake after EV treatment were not statistically significant. Discussing why this might be the case and whether a larger sample size or different assay conditions might yield statistically significant results would be valuable.

5.       Consider addressing the limitations of the 3D-MOTIVE chip model. For instance, discuss its ability to fully replicate the complexity of the in vivo renal microenvironment and whether it accounts for all relevant factors influencing nephrotoxicity.

6.       Suggest avenues for further research, such as exploring the long-term effects of MSCs-derived EVs, the optimal dosing regimen, and potential side effects or safety concerns that should be investigated in future studies.

7.       Consider discussing the translational potential of the findings, including the challenges and regulatory considerations associated with using MSCs-derived EVs in human clinical trials.

Comments on the Quality of English Language

Ok 

Reviewer 3 Report

Comments and Suggestions for Authors

This manuscript mainly evaluated the therapeutic effect of MSCs-derived EVs in cisplatin nephrotoxicity using a three-dimensional gravity-driven two-layer tubule-on-a-chip (3D-MOTIVE chip). The results showed in the 3D-MOTIVE chip, cisplatin 10μM decreased attached cells, albumin uptake, and tetra-methyl rhodamine methyl ester (TMRM) compared to the vehicle. Conversely, Annexin V and reactive oxygen species (ROS) were increased. After EVs at 4 and 8 μm/ml were used for treatment, cell viability was increased 2.8-fold and 2.5-fold, respectively, compared to the cisplatin-induced 20 nephrotoxicity group. Cell attachment was also increased 2.25-fold by EVs at 4 μm/ml and 2.02-fold by EVs at 8 μm/ml. TMRM and albumin uptake was also increased after EV treatment, although such increases were not statistically significant. Annexin V and ROS levels were decreased compared to those in the cisplatin-induced nephrotoxicity group. There were no significant differences in Annexin V and ROS levels according to EV concentration. These results could be of interest for researchers and doctors. However, there are still many problems for publication on its current form.

1.     The article has many grammatical errors, which needs polishing.

2.     For the concentration of EVs, why are only 4 and 8 μm/ml considered in this study?

3.     The part of evaluating cell viability in Figure 2 should be supplemented by a set of CCK-8 experiments.

4.     The authors have constructed very good composites, however, appropriate supplementation of cells and in vivo experiments can better reflect the advantages of the material.

5.     The reviewer suggests that the following articles in exosomes or extracellular vesicles could be a good reference for author to improve the manuscript:

(1)   Theranostics. 2022; 12(15): 6576-6594. doi: 10.7150/thno.78034

(2)   Cell Rep Med 2023 Jan 17;4(1):100881. doi: 10.1016/j.xcrm.2022.100881.

Comments on the Quality of English Language

Moderate editing of English language required

Author Response

We would like to thank you for the letter dated 05/10/2023, and the opportunity to resubmit a revised copy of this manuscript. We would also like to take this opportunity to express our thanks to the reviewers for the positive feedback and helpful comments for correction or modification. We believe have resulted in an improved revised manuscript, which you will find uploaded alongside this document. The manuscript has been revised to address the reviewer's comments, which are appended alongside our responses to this letter.

Q1. The article has many grammatical errors, which needs polishing.

A1. Thank you for pointing this out. I will attach the certificate after proofreading.

Q2. For the concentration of EVs, why are only 4 and 8 μm/ml considered in this study?

A2. Thank you for your question.

With preliminary data, cell viability, and albumin uptake tests were performed in 2D RPTEC cultures at each EV concentration (1 ug/mL and 5ug/mL). Increased viability and albumin uptake followed by cisplatin treatment were confirmed at 5 ug/mL EV, so experiments were performed at 4 ug/mL and 8 ug/mL on the 3D-MOTIVE chips.

Q3.The part of evaluating cell viability in Figure 2 should be supplemented by a set of CCK-8 experiments.

A3. Thanks for your advice.

The “Guidelines for cell viability assays” (doi.org/10.1002/fft2.44) agree that both CCK-8 and Calcein-AM can be used to measure cell viability. However, CCK-8 can be influenced by changes in intracellular metabolic activity, which do not directly impact cell viability. We referred to the paper titled "Multi-Organs-on-Chips for Testing Small-Molecule Drugs: Challenges and Perspectives" (doi.org/10.3390/pharmaceutics13101657), where cell viability was also assessed using Calcein-AM. Therefore, our research team decided that Calcein-AM would be a more suitable method.

Q4.The authors have constructed very good composites, however, appropriate supplementation of cells and in vivo experiments can better reflect the advantages of the material.

A4. I agree with you to some extent.

However, performing in vivo experiments is time-consuming, and a larger quantity of EVs is required in animal experiments to match the concentration of EVs found in the blood. Moreover, animal models present a limitation in that they cannot independently assess the effects on the proximal tubule, distal tubule, and glomerulus. Therefore, I aimed to highlight the advantages of the 3D-MOTIVE chip as a valuable tool for preclinical trials.

This content is described in line 211 of the Discussion section.

“In vivo models offer the advantage of observing functional changes in the entire kidney, but they have limitations in examining kidney structures in fine detail. Conversely, organ-on-a-chip technology has the flexibility to simulate either a distal tubule or glomerulus on the chip, depending on the specific research objectives.”

 Q5.    The reviewer suggests that the following articles in exosomes or extracellular vesicles could be a good reference for author to improve the manuscript:

(1)   Theranostics. 2022; 12(15): 6576-6594. doi: 10.7150/thno.78034

(2)   Cell Rep Med 2023 Jan 17;4(1):100881. doi: 10.1016/j.xcrm.2022.100881.

A5. Thank you for introducing good papers on extracellular vesicle research. I will refer to the papers you introduced to develop the paper.

We very much hope the revised manuscript is accepted for publication in the Journal.

Thank you for your consideration. I look forward to hearing from you.

Round 2

Reviewer 2 Report

Comments and Suggestions for Authors

Accepted in Current Form 

Reviewer 3 Report

Comments and Suggestions for Authors

Thank you for your reply. I have accepted your reply and think this article can be accepted.

Comments on the Quality of English Language

Minor editing of English language required